# From Traditional to Novel Printed Electrochromic Devices: Material, Structure and Device

**DOI:** 10.3390/membranes12111039

**Published:** 2022-10-25

**Authors:** Qingyue Cai, Haoyang Yan, Rihui Yao, Dongxiang Luo, Muyun Li, Jinyao Zhong, Yuexin Yang, Tian Qiu, Honglong Ning, Junbiao Peng

**Affiliations:** 1State Key Laboratory of Luminescent Materials and Devices, Institute of Polymer Optoelectronic Materials and Devices, South China University of Technology, Guangzhou 510640, China; 2School of Chemistry and Chemical Engineering, Institute of Clean Energy and Materials, Guangzhou Key Laboratory for Clean Energy and Materials, Huangpu Hydrogen Innovation Center, Guangzhou University, Guangzhou 510006, China; 3Department of Intelligent Manufacturing, Wuyi University, Jiangmen 529020, China

**Keywords:** ink formulas, printed electronics, optical properties, low cost, high performance, integrated molding

## Abstract

Electrochromic materials have been considered as a new way to achieve energy savings in the building sector due to their potential applications in smart windows, cars, aircrafts, etc. However, the high cost of manufacturing ECDs using the conventional manufacturing methods has limited its commercialization. It is the advantages of low cost as well as resource saving, green environment protection, flexibility and large area production that make printing electronic technology fit for manufacturing electrochromic devices. This paper reviews the progress of research on printed electrochromic devices (ECDs), detailing the preparation of ECDs by screen printing, inkjet printing and 3D printing, using the scientific properties of discrete definition printing method. Up to now, screen printing holds the largest share in the electrochromic industry due to its low cost and large ink output nature, which makes it suitable especially for printing on large surfaces. Though inkjet printing has the advantages of high precision and the highest coloration efficiency (CE) can be up to 542 ± 10 cm^2^C^–1^, it has developed smoothly, and has not shown rigid needs. Inkjet printing is suitable for the personalized printing production of high precision and small batch electronic devices. Since 3D printing is a new manufacturing technology in the 21st century, with the characteristics of integrated molding and being highly controllable, which make it suitable for customized printing of complex devices, such as all kinds of sensors, it has gained increasing attention in the past decade. Finally, the possibility of combining screen printing with inkjet printing to produce high performance ECDs is discussed.

## 1. Introduction

The world has entered the era of the fourth industrial revolution led by low carbon, environmental protection, etc. China has also put forward the development goal of a “double carbon economy”. With the social demand for energy saving in industrial production, how to effectively save energy and achieve sustainable development is currently an important research topic in the field of science and technology, so the innovation of technology is imperative, and the use of new energy sources is increasing. Electrochromic technology provides a new way for industrial energy saving [1,2,3,4,5,6]. Taking the application of building doors and windows as an example, the use of electrochromic technology for building doors and windows can selectively absorb or reflect external heat radiation and internal heat diffusion, which can reduce building energy consumption and protect the environment. It can be seen that the application of electrochromic technology plays a vital and positive role in the development of green building [7,8,9,10,11,12,13,14,15,16]. Electrochromic technology is widely used not only for building doors and windows but also for car, aircraft and ship [17,18,19,20,21]. According to research reported by David R. Roberts of the Renewable Energy Laboratory in the United States, homes with electrochromic smart windows consumed 9.1% less total energy and 13.5% less electricity in the home compared to homes with low-energy glass and shading glass [13]. Depending on the specific applications, ECDs work in the two ways: reflection and transmission. The former is applicable to EC displays and the latter can be applied in smart windows [22].

Printed electronics is a new process technology that applies traditional printing (or coating) technology to the manufacture of electronic components and products. Comparing with the conventional technologies, such as drop-casting, spray coating or spin, which are deficient in terms of reduced paint usage, deposition on large substrates and improved accuracy, printed electrochromic device offers the advantages of low cost as well as resource saving, flexibility and large area production. The two important components of printed electronic technology are printing method and conductive ink [23,24,25,26]. There are three commonly used printing processes in the previous phase, screen printing, inkjet printing, and 3D printing. The distribution of patents published in the web of SCI for these three printing methods over the years is shown in Figure 1. It can be seen that the use of printing method to manufacture electrochromic device has become popular years by years. Screen printing accounts for the largest portion of the market. Inkjet printing has developed smoothly and not showing rigid demand, but it has become the second most researched printing method after screen printing. Since 3D printing is a new manufacturing technology for the 21st century, it has gained increasing attention in the past decade.

## 2. Printed Electrochromic Device

With the development of printing electronic technology, it attracts more research institutions to apply printing method to electrochromic field. This paper starts with the introduction of a typical structure of electrochromic device, then describe the materials generally used, followed by screen printing, inkjet printing and 3D printing on transparent conductive layer, electrochromic layer and electrolyte layer. The modification of materials required for each printing method and the performance of the assembled devices are analyzed. Finally, we compare the pros and cons of different printing methods.

### 2.1. A Typical Structure of Electrochromic Device

A typical structure of electrochromic device is usually composed of five superimposed films, including substrate, two transparent conductive layers, an electrochromic layer, an electrolyte layer [17], as shown in Figure 2.

### 2.2. Printed Electrochromic Materials

From Figure 2, it can be seen that transparent conductive layers occupy two layers coated on glass or flexible substrate. When a voltage is applied, one layer acts as the electrode and the other as the counter electrode. It is required to have characteristics of high optical transparency, high conductivity and good compatibility. At present, common conductive materials are doped transparent conductive oxides, mainly ITO and FTO [27,28,29,30,31]. Poly(3,4-ethyldioxythiophene) (PEDOT) is not only a kind of electrochromic material but also a good transparent conductive polymer material. In addition, hybrid electrodes made of polyaniline (PANI) and metal nanowires incorporated into a polymer matrix are also a hot research topic [32,33,34,35]. The electrochromic layer is the core layer of the electrochromic device. The commonly used inorganic EC materials are WO_3_, NiO, V_2_O_5_, and organic EC materials are PEDOT, polyaniline, viologen, etc. [36,37,38,39,40]. Inorganic electrochromic materials are characterized by high coloration efficiency, cycle reversibility and chemical stability, while organic electrochromic materials are generally characterized by fast response time, high color tunability, and their electrical conductivity can be artificially adjusted by polymerization and doping of monomers [41,42,43,44]. The electrolyte layer, also known as ionic conductor layer, plays an important role in stabilizing the reaction voltage of electrochromic devices and avoiding irreversible electrochemical reactions caused by unreasonable voltage, it supplies counterions necessary to maintain the charge neutrality of electroactive materials during redox reactions [45,46,47]. Electrolyte materials can generally be classified as liquid electrolytes, solid electrolytes and gel electrolytes. The most commonly used liquid electrolytes are H_2_SO_4_ solutions and lithium salt plus organic solvents, such as propylene carbonate (PC) [48]. Solid electrolytes generally include fast ion conductor with inorganic salts containing Li^+^ and MgF_2_ doped with Li, etc. [49]. Gel state electrolytes are mainly added to liquid electrolyte polymers as a skeleton structure to conduct ions through the microporous structure of the polymer, [50,51]. Liquid electrolyte has the advantages of high conductivity, short response time, and good contact with electrochromic materials. However, liquid electrolytes are highly volatile and prone to leak, which increase the difficulty of encapsulating electrochromic devices, and safety and weathering issues limit theirs application. Although the pure solid electrolyte has the advantages of good mechanical properties, safety and no leakage problems, it is difficult to meet the practical application requirements of ECDs due to its low conductivity and interface contact problems with electrodes. Based on the advantages and disadvantages of both, gel electrolyte is a major research hotspot because it is solid with high conductivity, sufficient contact with the electrode interface [41,52,53,54].

### 2.3. Screen Printing Electrochromic Devices

#### 2.3.1. General Idea of the Screen Printing Process

Screen printing technology originated in ancient China as a method of transferring patterns onto fabrics more than 1000 years. Modern screen printing is a printing method in which the ink is transferred to the substrate through a patterned mesh by scraping and squeezing to form the same pattern on the screen, as shown in Figure 3a [55]. Screen painting requires inks to have high viscosity and low volatility [56]. Since polymers can meet this requirement, in most cases, researchers have either used polymer inks or added polymers as binders to nonorganic inks to modify the performances [57].

As one of the most well-established printing methods, screen printing is widely used in electrochromic devices and has the advantages of low production costs and simple operation compared with other film manufacturing techniques [58,59,60,61]. In addition, as shown in Figure 3b, the maximum printing speed can be increased to several meters per minute using the roll-to-roll printing process in rotary screen printing [62].

#### 2.3.2. Research on the Application of Screen Printing on Electrochromic Devices

There are two types of structure configuration for screen printed ECDs, namely the vertical structure and lateral structure, as shown in Figure 4 [63].

Figure 4 shows that in the vertical structure, electrode and EC layer, electrolyte layer and counter electrode are sequentially printed on the substrate. While in the lateral structure, the adjacent conductive layer and counter electrode are printed on the substrate first, then the electrolyte is transferred to them to bridge the gap. In the lateral structure, a lateral electric field is formed between the electrode and the counter electrode, and the ions move in the electric field over a distance that appears to be only the distance of the slit between the electrodes, but it is more difficult to achieve uniform coloring of the entire electrode because the ions need to traverse the entire electrode. In the vertical structure, on the other hand, the ions move at a distance of only the thickness of the electrolyte, and such a short distance of uniform transmission makes the electrochromic faster and the applied potential smaller. In addition, the filling factor of the vertical structure is higher because the counter electrode in the lateral structure requires a certain area. However, in the vertical structure, the counter electrode should be laminated to the surface of the electrolyte layer in order to reduce the current leakage caused by the electrolyte dissolving into the solvent of the top counter electrode. Since the lateral structure does not need this lamination technique, its manufacturing process is much simpler [58,62,63,64].

In reflection-type ECDs, both the vertical and the lateral structure are feasible. In the vertical structure, the substrate is usually opaque and the counter electrode is transparent. While in some cases, the substrate can be transparent and instead the counter electrode is opaque [58]. However, in transmission-type ECDs, the vertical structure is adopted in most occasions. In addition, each part of the ECD should be as totally transparent as possible to improve the performance of the transmittance spectrum [63,65].

1.Electrode

Traditional non-organic conductors, such as carbon and silver ink, can provide excellent conductivity and they have been widely applied in screen printing [58,66]. However, these kinds of ink are opaque therefore they are not suitable for certain situations where we need transparent electrode. To solve this problem, tin oxide materials such as indium tin oxide (ITO), antimony-doped tin oxide (ATO) and fluorine-doped tin oxide (FTO) were chosen to form screen-printed transparent electrodes. Nevertheless, due to the high cost of the preparation of the tin-oxide-based thin films, other transparent materials should be taken into consideration [26,67].

Poly(3,4-ethylenedioxythio-phene) doped with poly(styrene sulfonate) (PEDOT:PSS) is one of the most commonly used conductive polymers, as shown in Figure 5a [63]. Oxidized PEDOT is highly conductive but cannot dissolve in water, therefore insulating PSS is added to increase the solubility, rendering that the conductivity of PEDOT:PSS film is less than 1.0 S cm^−1^ [68]. Hence, to further improve its conductivity, one feasible method was decreasing the proportion of PSS. The level of PSS can be decreased by post-treatments of dipping or soaking PEDOT:PSS film in either organic solvents such as EG, DMSO and methanol (MeOH), or acids, such as high concentrated acetic acid and strong p-toluene sulfonic acid, then washing them off [63].

However, when fabricating electrochromic biosensors, the electrochromism of the PEDOT:PSS may preclude observation of the desired electrochromic contrast in several cases. Researchers commonly use Prussian Blue electrodes for electrochromic biosensors for the reason that it possesses both electrochromism and catalytic property for the reduction of oxygen and hydrogen peroxide [26,69].

2.Electrochromic layer

PEDOT:PSS can also use in electrochromic layer. It is nearly transparent to human eye in oxidized state and exhibits a dark blue in reduced state [55]. To increase the viscosity of ink, thickeners or gellants, polyvinyl alcohol (PVOH), carboxymethyl cellulose (CMC) or cellulose nanofibrils (CNF) can be added in PEDOT:PSS solutions, allowing screen printing to perform well [63].

Poly(3,4-propylenedioxythiophene)bis(ethylhexyloxy), referred to as ECP-Magenta, is a kind of poly(3,4-propylenedioxythiophene) derivatives, which is a novel screen-printable electrochromic material [66]. It exhibits an intense magenta color in the reduced and neutral state and nearly transparent in the oxidized state, with the electrochromic contrast over 60% [66]. To increase the conductivity, antimony-doped tin oxide microparticles (ATO/TiO_2_) were mixed in the ink. The final film showed significant electrochromic contrast at low potential (0.3 V vs. Ag/AgCl) in comparison with other electrochromic films and its electrochromic contrast was also relatively higher than others [22,66].

Since a great portion of EC materials are inorganic compounds, some of them can also be applied in screen printing. Prussian blue (PB) is one kind of screen-printable inorganic EC material which shows interesting electrochromic process. It can be reduced to Prussian White and oxidized to Berlin Green, as shown in [70,71]. To become qualified as screen-printable ink, metal oxide powder and polymer binder was added in the PB solution to improve the conductivity and viscosity, respectively [68,72]. However, the electrical resistance of the printed PB film is still high, therefore it is necessary to print it over a highly conducting surface [68]. Tungsten oxide is another kind of inorganic material which has been widely applied in smart window [65]. It exhibits excellent electrochromic property, as shown in Figure 6 [57]. To use screen printing process obtaining WO_3_ film, ammonium meta-tungstate (AMT) solution was chosen to work as the precursor solution. Then, polyvinyl alcohol (PVA) was added to improve the viscosity of the solution and the uniformity of the printed film. 2-Perfluoroalkyl ethanol (FSO) was also added in the AMT solution to decrease the surface tension and improve the wettability on the substrate. After screen printing the AMT solution, an annealing process was needed to remove PVA and produce WO_3_ [57].

3.Electrolyte layer

Since electrolytes used in EC devices are various, the kind of electrolyte ink can also be numerous as long as they meet the requirement for screen printing. The electrolyte material is usually transparent. However, to increase the electrochromic contrast in reflection-type devices, titanium dioxide (TiO_2_) can be added into the ink as the white pigment [58,62,64].

The most commonly used electrolyte in screen printable is gel since gel electrolyte has high viscosity, which is compatible with the screen printing process [66]. There are three sets of most commonly used polymetric gel electrolytes (PGEs), which are based on PMMA/PEO, PVB and PVDF, respectively. Some of them are shown in Figure 7 [65]. Each of these set has its own strengths and weaknesses. PVB-based and PVDF-based PGEs have higher stability and processability, while PMMA/PEO-based PGEs provide the best ionic conductivity and width of the electrochemical window. PDVF-based PGEs can also offer high ironic conductivity and wide electrochemical window as long as the electrode’s pore filling is good enough. In addition, PVB is inexpensive and readily available while PDVF membrane is far more scarce. Therefore, researchers should consider the trade-offs adequately when determining the kind of PGEs to use [65].

Despite the price, one of the most successful PGEs is poly(vinylidene fluoride-co-hexafluoropropylene) (P(VDF-co-HFP)) ionic liquid-based ion-gel, which is a type of PDVF-based PGEs. This gel has relatively high ionic conductivity, high transmittance (83%) and broad electrochemical window (3.2 V), which make it become outstanding [70].

4.Screen-printed Electrochromic Displays

Since screen printing has the feature of low-cost and large-area printing, screen-printed EC display is one of the research hotspots in recent years. The EC displays can be simply driven by passive-matrix, which has relatively uncomplicated structures and enables much easier manufacturing schemes [59,60,61]. However, since EC displays are especially susceptible to cross-talk, which is inevitable in the traditional passive-matrix addressing method, the active-matrix drive becomes popular. In this addressing method, each ECD is controlled by a corresponding transistor, hence the adjacent pixels can be considered as independent. The layout scheme of the electronic circuitry of the active-matrix EC display is shown in Figure 8a [73].

To fabricate the fully screen-printed display, the transistors should be prepared by screen printing, too. Similar to the ECDs, the transistors can also be printed in either the lateral structure (Figure 8b) or the vertical structure (Figure 8c) [62,64]. In comparison to the lateral structure, the vertical structure needs more processing steps but has faster switching time and smaller footprint. Moreover, to gain a desirable performance, future study may focus on further increasing the fill factor while reducing the pixel size [58,59,62,64].

To summarize, screen printing is a low-cost printing method for large areas and fast printing. Differing from other printing methods, it requires a higher ink viscosity, making polymers and gels ideal ink materials. Until now, researchers have been endeavoring to fabricate all screen-printed ECDs in either the vertical structure or the lateral structure. Future research may focus on increasing the ion conductivity of the ECDs and improving the driving methods.

### 2.4. Inkjet Printing Electrochromic Devices

#### 2.4.1. General Idea of the Inkjet Printing Process

Inkjet printing is a non-contact patterning technology that constitutes a personalized pattern by ejecting ink droplets to a specified location on the substrate. Inkjet technology requires tiny droplets of only 1 micron in diameter to be ejected from the nozzle in a matter of seconds. To ensure a clear composition of the image, the ink must print without blocking the nozzle and the substrate should form exactly the required ink dot size. Therefore, people need to rationalize the adaptation of the ink so that it has low surface tension, low viscosity, low relative density, fast drying speed, remains essentially neutral and avoids corrosion of the nozzle. According to the principle of inkjet printing, there are usually two modes, namely, continuous inkjet mode(CIJ, continuous inkjet with ink recovery storage device) and drop-on-demand mode (DOD, pulsed when the ink is needed to jet). The schematic diagram is shown in Figure 9 [74]. DOD method is widely promoted due to its high precision, controllability and low cost. In the midst of the two, piezoelectric inkjet printing can deposit patterns on a wide range of ink coverage, such as water, various organic/inorganic system solutions or UV inks, so piezoelectric DOD is commonly used for research and industrial purposes [75,76,77,78,79].

#### 2.4.2. Research on the Application of Inkjet Printing on Electrochromic Devices

Electrode

Conductive organic inks used in inkjet printing have the characteristics of softness and good compatibility with polymer substrates, but the conductivity is not ideal enough. Metal-based inks have the advantages of high conductivity and thermal conductivity, but their flexibility and scalability are insufficient. In 2007, William R reported the synthesis and inkjet process of water-dispersed polyaniline composites with a high loading rate of carbon nanotubes [80]. The printed films showed good optical transparency, sheet resistance and electrochromic properties, indicating that conductive active polymers and conductive carbon nanotubes can be prepared as conductive layers by inkjet printing technology [80]. Chiolerio developed acrylic conductive inks for inkjet printing in which conductive silver nanoparticles were generated in situ during UV-induced polymerization of acrylic formulations [34]. Kinner et al. used inkjet printing to create flexible transparent conducting electrodes based on silver nanowires. In their study, embedded and non-embedded silver grids were used to prepare the polymer light-emitting diode (PLED) as shown in Figure 10 [81]. During manufacturing, silver grids were printed in a honeycomb layout by inkjet and then coated with highly conductive PEDOT:PSS. Finally, to further improve the efficiency of the device, a honeycomb silver gate was embedded in the OrmocerVR-based material, which reduced the leakage current and enhanced the optical coupling. The results showed that the ink-jet printing efficiency of embedded AG-PEDOT:PSSPLED without ITO was significantly increased by 250% compared with ITO-based PLED [81].

2.Electrochromic layer

Inkjet printing has the characteristics of small ink droplets and high printing accuracy, so material applied to electrochromic layers can print patterns with high resolution and reduce material loss. Among all electrochromic materials, nanostructured WO_3_ is the most widely studied material [36,37,82,83,84,85]. However, WO_3_ nanoparticles may form many grain boundaries, so the resistance of the film can be very large. To solve this problem, Chang-Jian used conductive polymer PEDOT:PSS for surfactant and adhesive, making pure WO_3_ powder miniatures and dispersions [37]. Due to the polymer properties of PEDOT:PSS, the composite film prepared by inkjet printing had no crack while the surface is even. PEDOT:PSS can also act as a conductive adhesive to facilitate electron transport between WO_3_ nanoparticles. This strategy is not only applicable to WO_3_, but also suitable to other inorganic materials [37].

Compared with inorganic electrochromic material, conductive polymer materials such as viologen have the advantages of richer color, higher optical contrast, faster response and lower voltage control, so they have been widely studied by scientists [41,86]. In addition to synthesizing polymer to achieve multiple color, ligand/ion interactions in coordinated polymers can also be used to synthesize EC materials with different absorption spectra. Higuchi et al. synthesized a new class of EC materials, namely metal supramolecular polymers (MEPE), which exhibited electrochromic behavior based on REDOX reactions of inserted metal ions [87,88]. Chen printed a metal supramolecular polymer solution with two primary colors on a flexible electrode by inkjet printing [89]. By digitally controlling the printing dose of Ru-mepe ink and Fe-mepe ink, the color of printed EC film patterns can be adjusted directly without pre-mixing or synthesis of new materials, and ECD showed great contrast, electrochromic contrast (δt) of 40.1%, and coloring efficiency up to 445 cm^2^C^−1^, with a response time of 2 s [89], demonstrating the feasibility of fabricating display devices with different color settings through a full solution process [89].

Talking about high performance film formation, Pietsch reported on inkjet-printed dual mode devices based on the synchronous electrochromic and electrofluorochromic effect of the polyindenofluoren-8-tryarylamine polymer (PIF8-TAA) as shown in Figure 11 [90]. This material is highly transparent in its neutral state and changes to a deep red color by oxidation. Reference devices showed a high coloration efficiency of 542 ± 10 cm^2^C^−1^ at 395 nm [90].

However, manufacturing conjugated organic polymer based ECD with excellent electrochromic properties, custom motifs, and flexibility remains challenging. Huang reported inkjet printing, flexible and patterned ECD based on 2D polyaniline sheets uniformly dispersed in formic acid (FA), 2D layered polyaniline inks were prepared [32]. The device had a high optical contrast (76% at 750 nm) and a good coloring efficiency (CE) of 259.1 cm^2^C^−1^, short coloring/bleaching time (1.8/2.4 s), with false capacitance and mechanical flexibility [32]. In view of the current energy and environmental problems, more and more electronic products based on biofriendly materials are used in manufacturing. Pietsch had proposed a fully printed electrochromic display that was biodegradable to the international standard ISO14855 [91]. The electrochromic device was based on a biocompatible PEDOT:PSS active layer with gold electrodes and a biodegradable gel-based electrolyte printed on cellulose diacetate (CA) matrix [91].

3.Electrolyte layer

Because the electrolyte layer used by inkjet printing is less reported, we can only find a couple of applications. Lithium salt can be normally seen in liquid electrolyte with the advantages of good solubility, high ion conduction ability and ion migration number. Marques et al. invented a patent about using inkjet printing technique in electrolyte solution for electrochromic cell in which the electrolyte solution comprised lithium perchlorate, lithium triflate, lithium tetrafluoroborate or lithium hexafluorophosphate plus organic solvent [92]. Kondo used tetra butyl ammonium perchlorate as liquid electrolyte, fabricating a segment matrix EC display work in −0.5 V~1.5 V, switching between transparent and blue [93].

In short, the key to a successful inkjet printing device is the viscosity and surface tension of the ink. The viscosity of the ink should be appropriately low so that the ink can be ejected and can be replenished quickly. Meanwhile, the surface tension must be low enough to allow the ink droplets to globule out of the nozzle, therefore, requiring the prepared material to be significantly granular. Inkjet printing has the advantage of unlimited substrate, low cost, high precision, allowing for quantitative release of material without excess waste, which is advantageous in terms of resource conservation. However, when printing on a large area, it is easy to appear uneven printing, forming cracks and coffee rings, and irregular printing lines. Inkjet printing offers a wider selection of electrochromic materials than screen printing. Since the low viscosity inks usually make printed films smooth, inkjet printing is usually used in the manufacturing of electronics and optoelectronics such as transistors, electronic circuits, light-emitting diodes, and solar cells. Digital and custom patterns can be created on a myriad of substrates, including flexible films, to meet the personalized pattern design of electrochromic devices.

### 2.5. 3D Printing Electrochromic Devices

#### 2.5.1. General Idea of the 3D Printing Process

The basic principle is digital layering-physical lamination, which means that a digital model of the printed object is first established, digital layering is performed, and a two-dimensional processing path or trajectory is obtained for each layer. Then, choosing the appropriate material and the corresponding process, driven by aforementioned digital path, printing layer by layer, and finally cumulatively manufacture the product. The process is showed in Figure 12 [94]. The key points for 3D printing technology which differs from inkjet printing are 3D modeling with computer and equipment that supports layered printing. The equipment not only contains a nozzle to print dot, line, but most importantly to print layer by layer in its working cylinder through the removement of Z-axis. 3D printing is an evolving processing method that breaks away from traditional cutting processing and offers high processing flexibility and fast market response. With the advance of rapid prototyping technology, a dozen of 3D printing technologies applied in electrochromic devices are gradually developed such as aerosol jet-printed, fused deposition forming, direct ink writing, inkjet [95,96,97,98].

#### 2.5.2. Research on the Application of 3D Printing on Electrochromic Devices

Electrode

The electrode materials used in the above two printing methods can also be used in 3D printing as long as the ink formulas fit. Carbon nanotubes was used as electrode material to enhance the electrochromic contrast of the bicolor printed EC pixel benefited from its high transparency in the circuit designed by Ha [97]. Sundaram prepared a foldable electronic device which contained electrochromic element using silver nanoparticles [98]. Electrodes were laid on pins in the fold, and the device had a significant color transformation between −1 V and 1 V. Meanwhile, he used the same electrode material in another device—a strain sensitive sensor [95].

2.Electrochromic layer

3D printing has the advantages of precise physical reproduction, integrated molding, multiscale structure and controllable clipping design, so it can be used in the printing of electrochromic devices to reduce the printing process and realize the diversity of designed devices. Ha 3D printed a pixel driver circuit, which work under −1 V power supply, in which the electrochromic pixel used P3HT and PEDOT:PSS as the electrochromic material, reversibly turning from reddish purple to blue, and the printed circuit showed good stability within 600 consecutive cycles [97]. 3D-printed electrochromic devices can also be used on sensors, Sundaram had fabricated a strain-sensitive sensor from a variety of materials, including a 1.5 VDC powered electrochromic element fabricated by low temperature 3D printing, which used PEDOT:PSS mixed dimethyl sulfoxide as an electrochromic layer and driving circuit ink as shown in Figure 13 [95]. The switching ratio of the driving circuit was close to 10^4^ and obvious blue changes can be observed when the applied voltage was between −0.5 V and 1.25 V, in addition, the absorbance change was greater than 0.3A U at 700 nm [95].

Lee prepared a 3D-printed personalized multifunctional eyewear with a transparent electrochromic ion gel composed of 1-ethyl-3-methylimidazolium bis(trifluoromethyl sulfonyl)imide([EMI][TFSI]), ethyl viologen bis-(hexafluorophosphate)([EV(PF_6_)_2_]), hydroquinone (HQ) and poly(vinylidene fluoride-co-hexafluoropropylene) [96]. The glasses used UV light as the sensing signal and had a built-in UV sensor. When the UV light reached the threshold, the built-in power supply output 2 V voltage to make the glasses change color. Its working diagram can be seen in Figure 14. The glasses had a light modulation amplitude of 40% in the UV band and a response time within 30 s [96].

3.Electrolyte layer

In the printed drive circuit made by Ha, the electrolyte composite of EC pixels was an ionic gel based on trifluoromethylsulfonamide ([EMI][TFSA]) and triblock copolymer poly(Styreny-b-methyl methacrylate b-styrene)(PS−PMMA−PS) [97]. Efficient switching at 1 Hz was experimentally demonstrated by using highly conductive ionic gels in aerosol jet-printed EC pixels, and the possibility of faster switching on the order of 100 Hz was discussed [97]. In the foldable electronic device and strain-sensitive sensor designed by Sundaram, potassium chloride in deionized (DI) water, mixed with glycerol was used as liquid electrolyte [95,98]. The electrolyte bridged the control terminal and the electrochromic pixel pattern (PEDOT:PSS), forming a transparent cap for the enclosed electrolyte as shown in Figure 13a [95].

3D printing is a new printing method, its additive manufacturing approach can realize the natural seamless of products. 3D printing has grown faster than screen printing and inkjet printing in the past few years, due to its personalized manufacturing advantages and increasing printing accuracy. Due to its high processing flexibility and fast market response, 3D printing has developed rapidly in customized electronic functional devices. However, due to the layer-by-layer manufacturing method, the molding accuracy is relatively low, and the device might not be tightly connected to each other and prone to layered structures. In theory, 3D printing can produce parts with almost infinite complex shapes, so it is widely used to manufacture functional electronic circuits.

## 3. Summary

These three printing methods have something in common in terms of the materials used, as the materials do not change their structural properties during the printing process. Through changing the solvents, linkers and additives to adjust the viscosity, surface tension, density and so on physical properties of the inks in order to suit different printing methods. From the survey, it can be seen that PEDOT doped with PSS is one of the most used materials either in electrode or electrochromic layer due to its excellent conductivity and electrochromic properties. In addition, silver ink is also widely used for electrode printing due to its excellent conductivity and light transmittance. Since gel electrolyte combines the advantages of liquid and solid electrolytes, it is the most commonly used material fit for all three printing methods.

Comparing these three printing methods, screen printing can be used for large area displays, such as mobile phone, tablet as well as circuit board, such as the soldered resistive layer of PCB. The main problem is how to balance the mesh number and the viscosity of ink to improve precision. As the larger the mesh count, the smaller the width of the printing line. However, the increase in ink viscosity leads to difficulties in printing line width, good film continuity and increased edge neatness of the pattern. Inkjet printing is usually used in the manufacture of electronics and optoelectronics. However, it is not dominant in mass printing, and the requirements for ink drop viscosity and deposition uniformity are high. Therefore, future study in electrochromic devices may focus on optimizing the multi-jet printing technology as well as preparing more stable inks to form uniform films. From this point of view, it seems that if inkjet printing is used for fine circuit printing and then screen printing is used to lay down large-area working electrodes, the advantages of both can be combined to accomplish the preparation of large area electrochromic devices. 3D printing is usually used in complex circuits where the electrochromic devices are part of it. Limited by the size and functionality of current devices, there are few reports on the use of powder ink to prepare large-scale industrial prototype electrochromic 3D printing devices.

In the future, biodegradable and flexible polymer conductive materials will flourish with the development of ecological and flexible printed electronic products. Traditional printing technology will continue to evolve toward “green, functional, three-dimensional, device” development. In general, the development of these printing methods in the electrochromic field is manifested not only in the preparation of materials but also in the structural innovation of manufactured devices. In the near future, large area, low cost, versatile electrochromic devices will surely be developed.

## Figures and Tables

**Figure 1 membranes-12-01039-f001:**
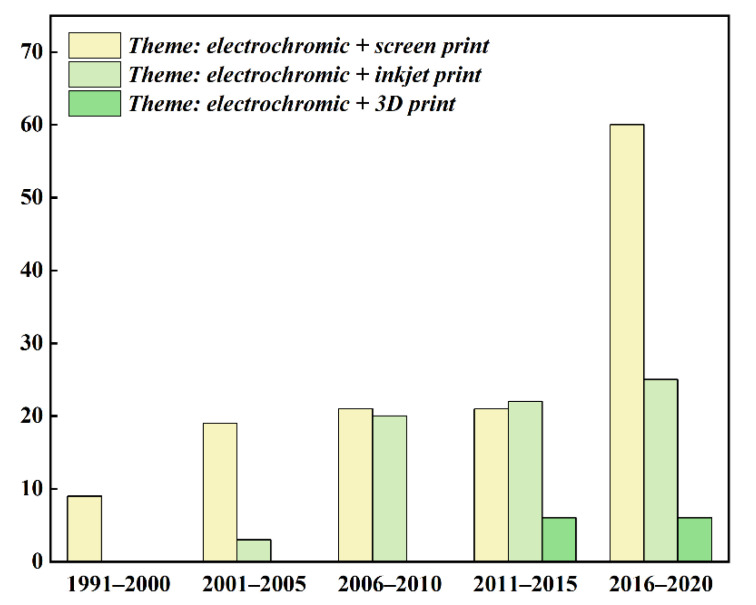
Distribution of patents for papers published in the web of SCI over the years for different printing methods of electrochromic coverage.

**Figure 2 membranes-12-01039-f002:**
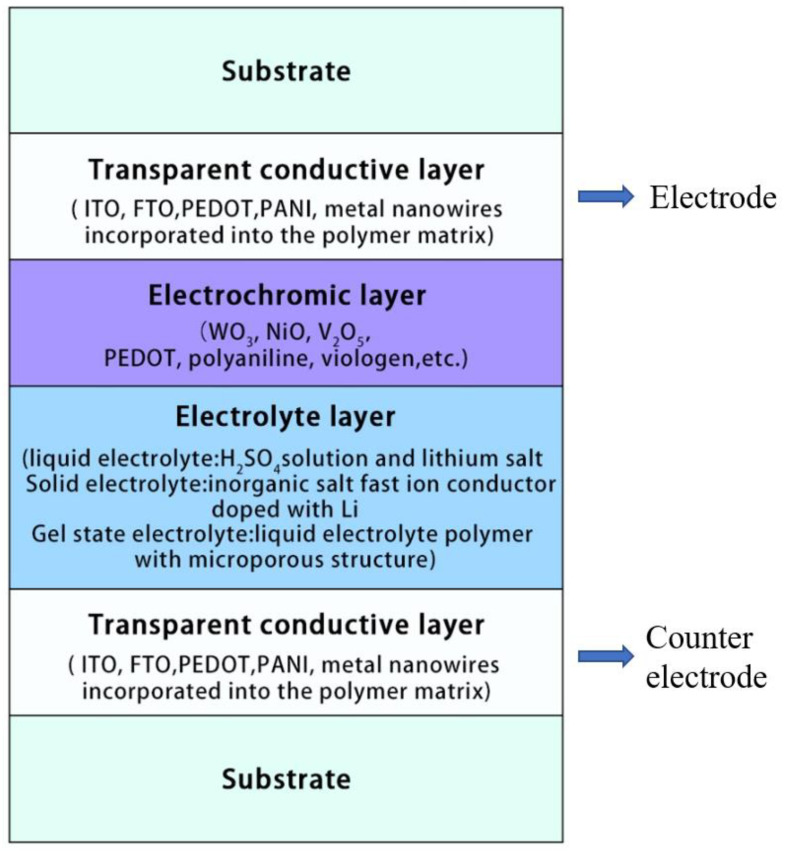
Configuration of a typical electrochromic device.

**Figure 3 membranes-12-01039-f003:**
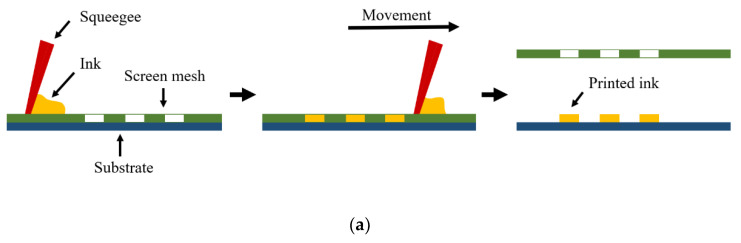
(**a**) Illustration of the screen-printing process [55]. (**b**) Schematic drawing of the roll-to-roll (R2R) screen printing process [62].

**Figure 4 membranes-12-01039-f004:**
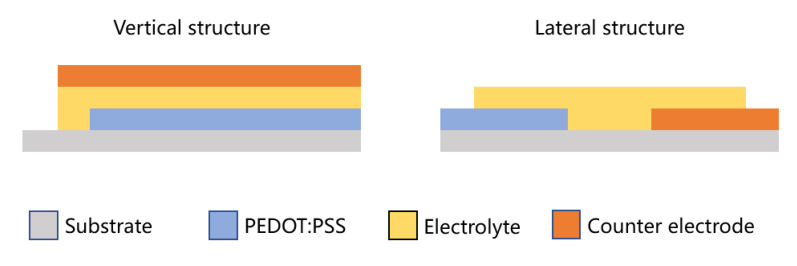
Schematic representation of the vertical and lateral structure of ECDs, using PEDOT:PSS as the EC material that can also serve as electrode. The counter electrode can also use PEDOT:PSS or other materials. Opaque materials such as PET are usually chosen for the substrate to improve the electrochromic contrast [63].

**Figure 5 membranes-12-01039-f005:**
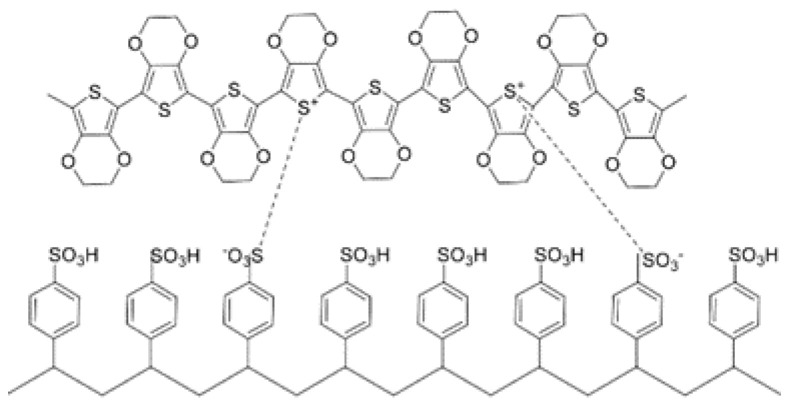
Molecular structure of PEDOT:PSS [63].

**Figure 6 membranes-12-01039-f006:**
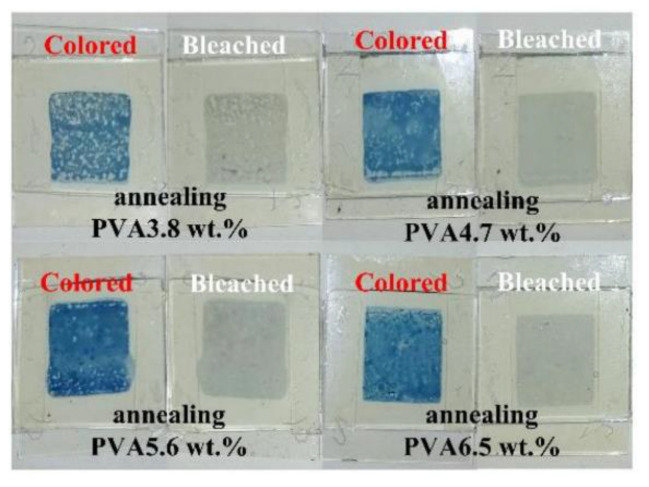
The photo images of screen-printed WO_3_ electrochromic devices using different ink formulas [57].

**Figure 7 membranes-12-01039-f007:**
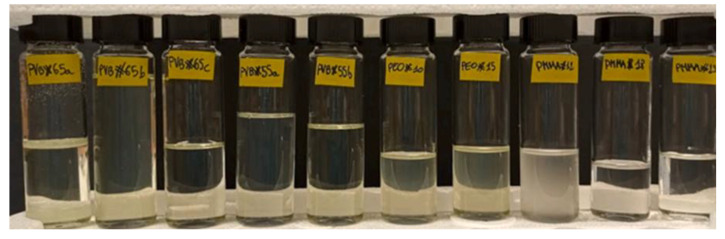
A representative picture of several PGEs considered in this study, from left to right are PVB#65a, PVB#65b, PVB#65c, PVB#55a, PVB#55b, PEO#10, PEO#15, PMMA#12, PMMA#18, PMMA#23 [65].

**Figure 8 membranes-12-01039-f008:**
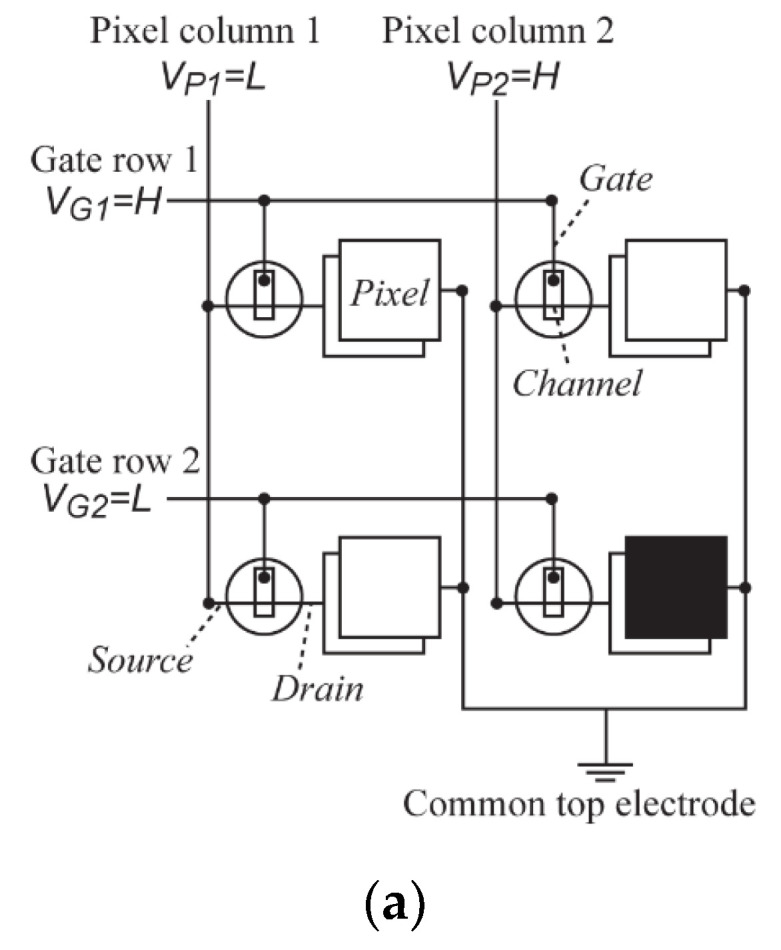
(**a**) Schematic drawing of the circuit layout of the active-matrix EC display [73]. (**b**) Top view of a laterally configured PEDOT:PSS-based depletion mode electrochemical transistor [62]. (**c**) Schematic architecture of a vertically configured organic electrochemical transistor device [64].

**Figure 9 membranes-12-01039-f009:**
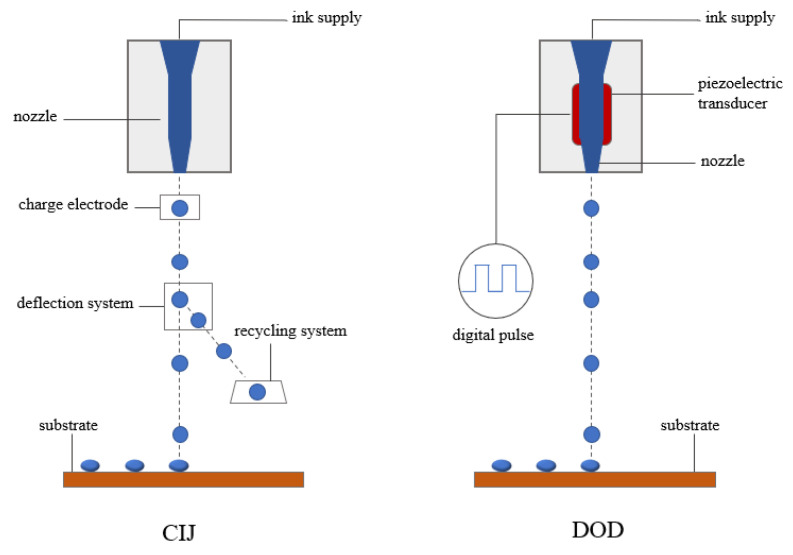
Schematic diagram of inkjet printing [74].

**Figure 10 membranes-12-01039-f010:**
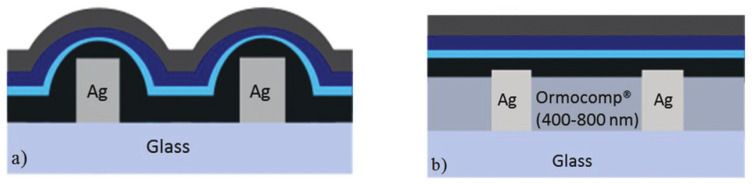
(**a**) Schematic representation of the PLED stack with non-embedded Ag-ink lines; (**b**) Schematic representation of PLED stack with embedded grid lines; [81].

**Figure 11 membranes-12-01039-f011:**
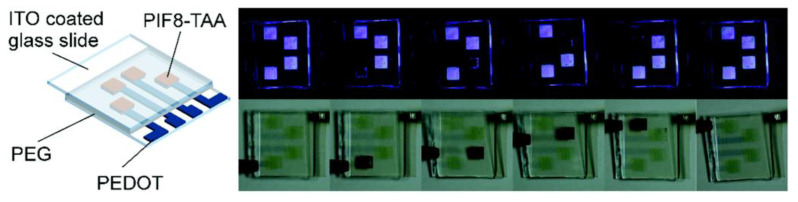
EFCDs with printed electrodes and electrofluorochromic layers under operation [90].

**Figure 12 membranes-12-01039-f012:**
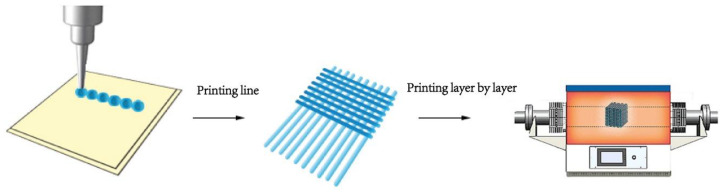
Schematic diagram of 3D printing process [94].

**Figure 13 membranes-12-01039-f013:**
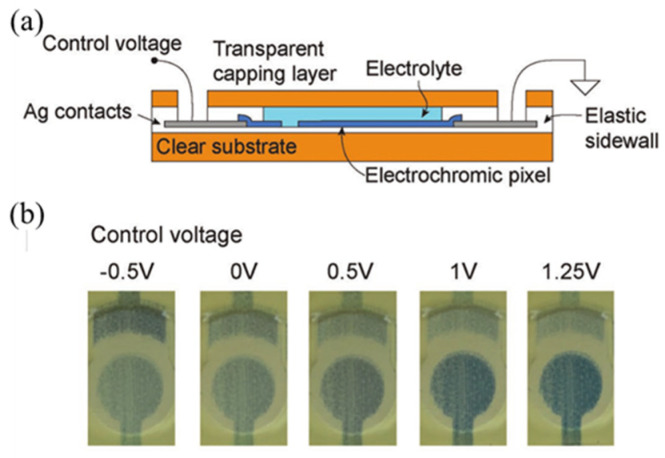
(**a**) Schematic cross-section of the electrochromic pixel. (**b**) The transmitted color through the sealed printed pixel placed above a white paper in natural light. The circular pixel is reduced as the control voltage is increased [95].

**Figure 14 membranes-12-01039-f014:**
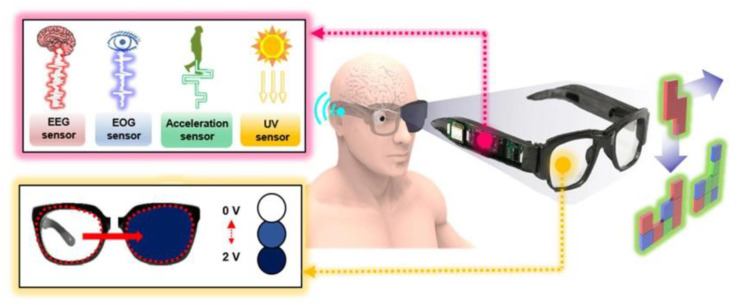
Working diagram of Smart Electronic Eyeglasses [96].

## Data Availability

The data that support the findings in this study are available from the corresponding author upon reasonable request.

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
