# Peer review of "From Traditional to Novel Printed Electrochromic Devices: Material, Structure and Device"

_membranes, 2022, doi:10.3390/membranes12111039_

Round 1
Reviewer 1 Report
In this present manuscript, Cai et al. reviewed the electrochromic devices fabricated by printed electrodes. The topic is very relevant in today's age and the manuscript is well-composed. I recommend the publication of the manuscript after a few modifications.
(1) Authors should discuss the advantages of printed electrodes over other technologies like drop-casting, spin, or spray coating used to fabricate the electrochromic film for the electrochromic devices.
(2) The image quality may be improved. Especially in Figures 2, 7, 14, etc. The write-up inside the figures is difficult to read.
Author Response
Dear reviewer,
Thanks for your opinions, these comments are very helpful to improve the quality of the manuscript. For details of the revisions, please see the attachment.

Reviewer 2 Report
The review entitled “From traditional to novel printed electrochromic devices: material, structure and device” describes different approaches to the fabrication of electrochromic devices - namely, screen-printing, inkjet printing and 3D-printing. Due to the growing climatic and energy issues, this is quite an important scientific field. The review summarized a variety of recently published papers, but unfortunately it suffers from a very vague structure and poor English. Therefore, I would recommend publishing it only after a very systematic text revision. Below one could find some selected points that require attention.
1. The text is full of grammatical mistakes, typos and confusing sentences. It’s highly recommended to check English throughout the paper.
2. Figure 1. It looks very strange, that total number of “electrochromic + print” papers surpasses a sum of other three columns (i.e. “electrochromic + screen print”, “electrochromic + inkjet print”, “electrochromic + 3D print”. You should clarify this point.
3. Current chapter's structure makes no sense at all. For instance, see “2. Printed electrochromic structure” and “3. Printing electrochromic devices”. What’s the difference? “3.1. Screen printing electrochromic devices” and “3.1.1. Screen printing technology in electrochromic” - exactly the same situation. Or another example - chapter 2.1 without 2.2, 2.3., etc. Seems that it's possible to mix different chapters and also name some of them as "General idea of ..."
4. Figure 2. Where’s the counter-electrode? It’s impossible to have only a redox reaction in electrochromic material. And also I should note that you mention counter-electrode in subsequent parts of text.
5. “Electrolyte 97 layer, also known as ionic conductor layer, plays an important role in stabilizing the reac-98 tion voltage of electrochromic devices and avoiding irreversible electrochemical reactions 99 caused by unreasonable voltage” (Lines 97 - 100). You didn't mention the main function of electrolyte: it supplies counterions necessary to maintain the charge neutrality of electroactive materials during redox reactions. It would also be helpful to add some recent papers discussing the essential role of counterions in the performance of electrochromic materials [doi.org/10.1016/j.solmat.2019.04.004 ;doi.org/10.1016/j.synthmet.2018.03.001;10.1039/D0RA01436B]
6. “Comparing with the lateral structure, the vertical structure has shorter carrier path length, rendering more rapid color switching response and smaller applied potential” (Line 146-148). Please, clarify the nature of this effect. It’s not obvious why vertical structure can ensure a shorter carrier path length.
7. At line 176 you mentioned the word "electrode", but you hadn't introduced it previously (see figure 2, there is no electrode there). I'm assuming you mean substrate + transparent conductive layer, but this should be clearly stated in the text.
8. Some information is repeated many times in the text. See lines 186-189, 170-76, 85-90, describing the possibility to use PEDOT-PSS. They are almost the same, except for minor differences.
9. Color change, Color contrast, Transmittance difference, electrochromic contrast. You are using a lot of different words that mean only one term - electrochromic contrast. This can be very confusing for the reader. It would be better to use one or two of them, no more.
10. Lines 300 - 302. How is it possible to use ITO or FTO as an ink? It's hard to imagine
11. The description of the 3D approach (Lines 400-405) is very brief and doesn't provide any explanation fot the difference from inkjet printing.
Author Response

(The authors gave the same response as above.)

Round 2
Reviewer 2 Report
Thank to the careful work of author their work now looks much more informative and accurate and could be published in Membranes.